# Genome-Wide Identification, Characterization and Expression Profiling of the *CONSTANS*-like Genes in Potato (*Solanum tuberosum* L.)

**DOI:** 10.3390/genes14061174

**Published:** 2023-05-28

**Authors:** Ruining Li, Ting Li, Xiang Wu, Xuyang Yao, Hao Ai, Yingjie Zhang, Zhicheng Gan, Xianzhong Huang

**Affiliations:** 1Center for Crop Biotechnology, College of Agriculture, Anhui Science and Technology University, Chuzhou 233100, China; 2College of Life Sciences, Shihezi University, Shihezi 832003, China

**Keywords:** *S. tuberosum*, genome-wide analysis, *COL*, tuber, gene expression

## Abstract

*CONSTANS*-like (*COL*) genes play important regulatory roles in flowering, tuber formation and the development of the potato (*Solanum tuberosum* L.). However, the *COL* gene family in *S. tuberosum* has not been systematically identified, restricting our knowledge of the function of these genes in *S. tuberosum*. In our study, we identified 14 *COL* genes, which were unequally distributed among eight chromosomes. These genes were classified into three groups based on differences in gene structure characteristics. The COL proteins of *S. tuberosum* and *Solanum lycopersicum* were closely related and showed high levels of similarity in a phylogenetic tree. Gene and protein structure analysis revealed similarities in the exon–intron structure and length, as well as the motif structure of COL proteins in the same subgroup. We identified 17 orthologous *COL* gene pairs between *S*. *tuberosum* and *S. lycopersicum*. Selection pressure analysis showed that the evolution rate of *COL* homologs is controlled by purification selection in *Arabidopsis*, *S*. *tuberosum* and *S. lycopersicum*. *StCOL* genes showed different tissue-specific expression patterns. *StCOL5* and *StCOL8* were highly expressed specifically in the leaves of plantlets. *StCOL6*, *StCOL10* and *StCOL14* were highly expressed in flowers. Tissue-specific expression characteristics suggest a functional differentiation of *StCOL* genes during evolution. Cis-element analysis revealed that the *StCOL* promoters contain several regulatory elements for hormone, light and stress signals. Our results provide a theoretical basis for the understanding of the in-depth mechanism of *COL* genes in regulating the flowering time and tuber development in *S. tuberosum*.

## 1. Introduction

The photoperiod regulates not only flowering but also different developmental transitions, including flower bud differentiation, seasonal growth cessation in trees and tuber formation and development [1,2]. Tuber formation and plant flowering are regulated by the photoperiod. The leaves sense day length to induce flowering and tuber formation, and the signals produced have been named florigen and tuberigen, respectively [3,4]. Early studies revealed that short-day (SD) conditions can induce the tubers of *Solanum tuberosum* [3,5,6]. However, the regulation of tuber formation and development by short-day conditions requires the participation of the *CONSTANS* (*CO*) and *COL* gene families [7,8,9].

COL is a zinc-finger transcription factor that plays an important role in various biological processes, such as flowering time control, regulation of growth and development and stress responses [10,11,12]. The COL protein contains two conserved domains: a B-box domain at the N-terminus and a CO, COL, TOC 1 (CCT) domain at the C-terminus [13,14,15]. The B-box domain is mainly involved in regulating protein interactions [16], while the CCT domain is mainly involved in the nuclear localization of the COL protein, thereby regulating gene transcription and nuclear protein transport [17]. The structure of the B-box and CCT affect the function of *COL* genes, and the mutation of the B-box domain may lead to *COL* gene loss of function [18].

*CO* gene expression is consistent with light rhythms, which promotes flowering only under long-day (LD) conditions [19,20]. *FLOWERING LOCUS T* (*FT*) is the florigen gene that researchers had long sought [21,22], and *FT* is a direct downstream gene of the transcription factor CO regulating plant flowering. A conserved zinc finger domain in the CO protein can directly bind to the *FT* promoter to regulate expression of the *FT* gene [19,23,24]. CO protein activates the *FT* expression and *SUPPRESSOR OF OVEREXPRESSION OF CO 1* (*SOC1*) under LD conditions [25]. Under SD conditions, the *CO* homolog *Heading Date 1* (*Hd1*) promotes the expression of *FT* in rice, while the *FT* homolog *Hd3* is inhibited under LD conditions [26,27]. In *Arabidopsis*, CO induces the transcription of *FT* and *TWIN SISTER OF FT* (*TSF*) in the late afternoon under LD conditions [28,29]. The flowering time controlled by the *CO*/*FT* module is highly conserved in photoperiod-sensitive plants, while this conservation is inconsistent in different species [30,31,32]. *AtCOL5* promotes flowering in *Arabidopsis* [33]. *AtCOL3* regulates light morphogenesis positively and is involved in anthocyanin accumulation and root development [34]. *AtCOL7* regulates starch biosynthesis in leaves [35]. The expression of *AtCOL8* is observed in seeds, leaves, flowers and siliques, and *AtCOL9* is involved in the regulation of flowering time by reducing the expression of *CO* and *FT* [36]. In rice, *OsCOL10*, *OsCOL13* and *OsCOL16* play negative regulatory roles under LD and SD conditions, *Hd1* promotes flowering under SD conditions and inhibits flowering under LD conditions [10,37,38,39].

Tuber formation in transgenic potato plants heterologously expressing *AtCO* occurs more than 7 weeks later than that of wild-type plants under SD conditions [7]. Grafting experiments demonstrated that grafting the upper part of wild-type potato plants on to potato plants heterologously expressing *AtCO* does not affect the late formation of potato tubers. By contrast, when potato plants heterologously expressing *AtCO* are grafted on to wild-type potato plants, the potato tuber formation is significantly delayed [7,40]. *AtCO* in plant leaves therefore plays a negative regulatory role during tuber formation and development in the potato. Gonzalez-Schain et al. cloned *StCO* in potato, a homolog of *AtCO*, and found that *StCO* and *AtCO* are functionally similar [40]. The plants of potato overexpressing *StCO* form tubers later than wild-type plants under low photoperiod induction conditions. *StCO* silencing promotes tuber formation under photoperiod inhibition and low photoperiod induction conditions, but has no effect under strong SD induction conditions, indicating that *StCO* inhibits tuber formation in a photoperiod-dependent manner [8]. In addition, potato StCOL1 protein inhibits tuber formation by activating the expression of the *FT* homolog *StSP5G* [9]. The silencing of *StCOL1* or *StSP5G* leads to the activation of *StSP6A* in leaves, and negative regulation of tuber development by StCOL1 is dependent on *StSP5G* [9]. An additional level of control among the expression of *StCOL1*, *StSP5G* and *StSP6A* determines the plant tuberization transition [41,42]. In conclusion, the *COL* gene family plays an important regulatory role in the formation and development of potato tubers.

As a nutrient storage organ of potato, the formation of tubers is key for the survival and reproduction of potato plants and reflects the economic value of potato crops. Research on the mechanism of tuber formation and development is not only an important part of potato developmental biology research, but also an important guarantee for improving the yield and quality of potato. Potato genome sequencing has been completed, and a series of breakthroughs has been made in genome research [43,44,45,46]. For example, they offered a holistic view of the genome organization of a clonally propagated diploid species [44], they employed genome design to develop a generation of pure and fertile potato lines and thereby the uniform, vigorous F_1_s [45], they assembled 44 high-quality diploid potato genomes from 24 wild and 20 cultivated accessions [46], facilitating the exploration of the functions of related genes in potato growth and development. Previous studies have shown that the *COL* gene family is involved in regulating tuber formation and development by sensing the photoperiod and circadian rhythm changes in potato. However, studies on the function and regulatory mechanism of potato *COL* genes involved in tuber development are still very limited, and there has been a lack of systematic genome-wide identification and functional analysis of potato *COL* gene family members. In this study, we used bioinformatics methods to identify *COL* gene family members from the whole potato genome, carried out BLAST searches and mapped the genes to their respective chromosomal positions, analyzed the evolutionary relationship between potato *COL* genes and genes from other plant species using multiple sequence alignment and phylogenetic tree construction, determined the structural characteristics of genes and proteins, examined gene selection pressure, contraction and expansion, and identified the *cis* acting elements of promoter regions. We also studied the expression specificity of potato *COL* genes in different tissues and at different development stages. This study provides a theoretical basis for further research on the functions of *COL* gene family members in tuber formation and the development of potato.

## 2. Materials and Methods

### 2.1. Plant Materials

Potato (*S. tuberosum*) plantlets (cv. *Favorita*) were provided by the Center for Crop Biotechnology, Anhui Science and Technology University. Potato plantlets were cut into 1–1.5 cm segments with leaves and then placed into medium in tissue culture bottles (inner diameter, 63 mm; height, 85 mm). Each bottle contained four stem segments and was placed in a tissue culture room with a relative humidity of 65 ± 5%, a daytime temperature of 22 ± 2 °C, a night temperature of 18 ± 2 °C, a photoperiod uniformly set to 8-h light/16-h dark and a photosynthetic photon flux density of 65 μmol m^–2^ s^–1^. The medium used for propagation of plantlets in vitro was solid Murashige and Skoog (MS) medium (4%, *w*/*v*, sucrose, 0.9%, *w*/*v*, agar). The medium used for the induction of microtubers was solid MS medium (8%, *w*/*v*, sucrose, 0.9%, *w*/*v*, agar). Potato plantlets grown for 30 days were acclimated and transplanted to pots containing vegetative soil and vermiculite (1:1) and placed in a plant growth room where the relative humidity was 65 ± 5%, the temperature was 22 ± 2 °C, and the photoperiod was uniformly set at 12-h light/12-h dark. The light intensity was 200 μmol m^–2^ s^–1^. The leaves, stem segments and shoots of the middle potato plantlets grown for 10 days and three microtubers per bottle grown for 80 days were collected under tissue culture room, performing three biological replicates. The leaves and stem segments of the middle plants, roots, tubers and flower tissues of plants grown for 70 days were collected under plant growth room, performing three biological replicates. These tissues were immediately flash-frozen in liquid nitrogen and stored at −80°C for RNA extraction and gene expression analysis.

### 2.2. Identification of COL Gene Family Members in S. tuberosum

The genome sequence, proteins and corresponding coding sequences of *S. tuberosum* were downloaded from the Phytozome v13 website (https://phytozome.jgi.doe.gov/pz/portal.html, accessed on 10 November 2022). An algorithm-based BLASTP was performed using the amino acid sequences of *A. thaliana* COL proteins as queries in the protein databases of *S. tuberosum*, *S. lycopersicum*, *Nicotiana tabacum* and *Oryza sativa*, with an E < 1 × 10^−5^ and other parameters as the default values. The conserved domain models of CCT (PF06203.15) and zinc finger B-box (PF00643.25) were downloaded from Pfam 33.1 (http://pfam.x fam.org, accessed on 10 November 2022) [47]. The conserved protein sequences of the *COL* gene family were obtained using HMMER3.0 software [48], with a threshold value of *E* < 10^−5^; domains of candidate COL protein sequences were identified using the SMART database (http://smart.embl-heidelberg.de, accessed on 10 November 2022). The sequences with both CCT and zinc finger B-box (ZF-B-box) domains were retained [13,49]. Finally, the proteins were named according to the gene annotations. The website Cell—Ploc 2.0 (http://www.csbo.sjtu.edu.cn/bioinf/Cell-PLoc-2, accessed on 15 November 2022) was used for predicting the subcellular localization of candidate genes encoding COL proteins. The online ExPASy (https://web.expasy.org/protparam/, accessed on 17 November 2022) was used to analyze the physicochemical properties of the amino acids encoded by the candidate genes.

### 2.3. Phylogenetic Tree Reconstruction, Analysis of Gene Structure and Protein Motifs

ClustalW [50] was used for the multiple sequence alignment of the potato COL family protein sequences with default parameters. COL family protein sequences of *A. thaliana*, *S. lycopersicum*, *N. tabacum* and *O. sativa* were download from NCBI (https://www.ncbi.nlm.nih.gov/, accessed on 25 November 2022), and the COL protein sequences of these species and potato were used to reconstruct a phylogenetic tree. The phylogenetic tree of the COL protein family members was reconstructed using MEGAX 10.1.8 software [51] and the neighbor-joining method, with the bootstrap value set to 1000 cycles. The gff3 file of the COL family was submitted to GSDS (http:/gsds.cbi.pku.edu.cn, accessed on 1 December 2022) for gene structure analysis [52]. The MEME (http://meme-suite.org/, accessed on 4 December 2022) website was used for the motif prediction with the number set to 10. Conserved domain structure was analyzed using batch CD-search (https://www.ncbi.nlm.nih.gov/Structure/bwrpsb/bwrpsb.cgi, accessed on 6 December 2022).

### 2.4. Chromosomal Location

TBtools v1.09876 software [53] was used to calculate the location of each gene and chromosome length information in the potato genome. The physical location of *StCOL* genes on the chromosomes was constructed using MG2C (http://mg2c.iask.in/mg2c_v2.0/, accessed on 6 December 2022).

### 2.5. COL Gene Selection Pressure and Duplication Type Analysis

Blast-2.13.0 + was used for comparison, and MCScanX [54] was used to calculate collinearity of homologous *COL* genes within and between species, with a threshold of *E* < 10^−5^. StCOL family members were imported into MEGAX [51] for multiple sequence alignment, and then DnaSP 6.0 software [55] was used to calculate the non-synonymous substitution rate (*Ka*) and synonymous substitution rate (*Ks*) of the orthologous genes. The *Ka*/*Ks* values of collinear gene pairs in *Arabidopsis*, *S. lycopersicum* and *N. tabacum* were used to evaluate the selection pressure on orthologous *COL* genes during evolution. *Ka*/*Ks* > 1, < 1 or = 1 indicates a positive, negative or neutral evolution, respectively [56]. TBtools v1.09876 software [53] was used to show the collinearity of *COL* among *Arabidopsis*, *S. lycopersicum*, *N. tabacum* and potato to judge the collinearity of the *COL* gene families.

### 2.6. Tissue Expression Characteristics of the StCOL Gene in S. tuberosum

RNA extraction and reverse transcription were performed according to Huang et al. [57]. Quantitative Real-Time PCR (qPCR) was performed according to Jin et al. [58]. PCR was performed using an ABIViiA7 real-time PCR instrument (Life Technologies, Carlsbad, CA, USA). The Primer 3.0 tool (https://bioinfo.ut.ee/, accessed on 15 December 2022) was used to design *StCOL* the gene-specific amplification primers; *elongation factor-1alpha* (*EF1α*) was used as a reference gene [59] (Appendix A). The sequence alignment between the primers by DNAMAN was used to verify the specificity of primers. In addition, the PCR amplicons were sequenced to verify the amplification of the *StCOL* genes. Three independent biological replicates of qPCR were performed, and each PCR reaction was performed in triplicate. The 2^–ΔCT^ method was used to calculate the relative expression levels of genes in different tissues [60]. GraphPad Prism 9.5.0 software was used for the statistical analysis and drawing of data.

### 2.7. Analysis of Cis-Elements in the Promoter Regions of StCOL Genes

The sequence spanning 2000 bp upstream of the initiation codon of each *StCOL* gene was submitted to PlantCARE (http://bioinformatics.psb.ugent.be/webtools/plantcare/html/, accessed on 16 December 2022) for predicting the cis-elements. TBtools v1.09876 software [53] was used to draw the *cis*-elements in the promoter region.

## 3. Results

### 3.1. Identification and Chromosome Distribution of COL Gene Family Members

A total of 14 *COL* genes were identified from the potato genome by genome-wide identification, they were conducted as per their location on the chromosomes (Table 1, Appendix A). Each of the 14 StCOL proteins encoded contained two conserved domains: a B-box domain at the N-terminus and a CCT domain at the C-terminus (Appendix A). The analysis of the amino acid characteristics of the StCOL family proteins showed that the StCOL proteins were composed of 347–453 amino acids; the StCOL11 amino acid sequence length was the shortest at 347 aa, while the StCOL6 amino acid sequence length was the longest at 453 aa. The molecular weight of the proteins ranged from 38,651.0 to 51,218.4 Da, with StCOL5 having the largest molecular weight (51,218.4 Da) and StCOL11 having the smallest molecular weight (38,651.0 Da). The theoretical isoelectric points ranged from 5.20 to 6.09, indicating acidic proteins. The protein localization prediction results showed that all the COL proteins in potato are localized in the nucleus (Table 1). In addition, blast-2.13.0 + analysis showed that the similarity between the *StCOL1* sequence and the *StCO* sequence studied by Gonzalez-Schain et al. was 98.148% [8], suggesting that *StCOL1* might be the *StCO* gene.

### 3.2. Phylogenetic Tree Analysis of COL Protein Families in Five Plant Species

To investigate the phylogenetic relationships of *COL* family genes, we retrieved the amino acid sequences of COL proteins from *S. tuberosum*, *S. lycopersicum*, *N. tabacum*, *O. sativa* and *A. thaliana* databases and reconstructed the phylogenetic trees using multiple sequence alignment. According to the topological structure of the phylogenetic tree, potato COL proteins could be divided into subgroups I, II and III (Figure 1 and Figure 2A). There were five COL proteins in subgroup I, three COL proteins in subgroup II and seven COL proteins in subgroup III. From the phylogenetic tree, it can be seen that the COL proteins of *S. tuberosum* and *S. lycopersicum* are closely related and have high similarity.

### 3.3. Phylogeny, Gene Structure and Protein Motifs

The evolutionary tree of 14 *COL* proteins in potato showed that these proteins cluster into three subgroups (Figure 2A). Gene structure analysis showed the *StCOL* genes contained 2–4 exons and 1–5 introns (Figure 2B). In subgroup I, the *StCOL13*, *StCOL11* and *StCOL9* genes contained two exons and one intron, and the *StCOL1* and *StCOL2* genes contained three exons and two introns. In subgroup II, all the seven genes contained four exons. In subgroup III, all three genes contained two exons and three introns. Although the length of genes in the same subgroup was different, the exon–intron length and the structure of genes in the same subgroup were similar. We next analyzed the motif distribution of COL proteins, and ten conserved motifs were predicted (Figure 2C). The length of the motifs ranged from 21–50 amino acids. The high similarity of the motif structure of the COL proteins in the same subgroup indicated that the *COL* gene family is highly conserved. However, some differences were also found. The subgroup I and III proteins had motif 1, while subgroup II proteins had not. The subgroup III proteins had motif 6 while subgroup I and II proteins had not. The subgroup I and II proteins had motif 3 while subgroup III proteins had not, and all COL proteins contained motif 2 and motif 4. Protein domain analysis showed that all members of the COL family had a CCT domain near the carboxyl terminus, with subgroup I and III proteins containing Bbox_SF1 and Bbox_SF2 and subgroup II proteins containing Bbox_SF1 (Figure 2D).

### 3.4. Chromosome Localization

Chromosomal mapping results showed the 14 *StCOL* genes were unevenly distributed on eight chromosomes (Appendix A), with 1–4 genes on each chromosome. Chr03, chr04, chr08 and chr09 each had one gene; chr02 and chr07 each had two genes; chr05 had four genes.

### 3.5. Selection Pressure on COL Genes and Contraction Versus Expansion

The analysis of the evolutionary selection pressure (*Ka*/*Ks*) of *COL* family genes in four species showed that the *Ka*/*Ks* of collinear gene pairs in *A. thaliana*, *S*. *tuberosum* and *S*. *lycopersicum* were all less than 1, indicating that the purification selection was the main factor in the evolution of *COL* genes. The *Ka*/*Ks* of collinearity gene pairs in *N*. *tabacum* was close to 1, suggesting a neutral selection (Figure 3). In order to study the contraction and expansion of *COL* gene family members during evolution, we analyzed the collinearity of orthologous *COL* genes among *A. thaliana*, *S*. *tuberosum*, *S*. *lycopersicum* and *N*. *tabacum* (Figure 4). There were 12, 12 and 8 orthologous genes of *StCOL* in *A. thaliana*, *S*. *lycopersicum* and *N*. *tabacum*, respectively. There were 11, 17 and 16 pairs of collinear genes between *S*. *tuberosum* and *A. thaliana*, *S*. *lycopersicum* and *N*. *tabacum COL* homolog, respectively. These results indicated that the *StCOL* family members have expanded. The analysis of paralogous *COL* gene pairs in the potato genome showed that *StCOL* has three paralogous gene pairs, namely *StCOL11*/*13*, *StCOL10*/*14* and *StCOL3*/*4*, with each gene located on a different chromosome (Appendix A).

### 3.6. Tissue Expression Characteristics of StCOL Genes

We next analyzed the expression characteristics of *StCOL* genes in nine tissues using qPCR. The expression levels of 14 *StCOL* genes were significantly different in different tissues of potato (Figure 5). The expression patterns of *StCOL1, StCOL11* and *StCOL13* were similar, with high expression levels in the leaves of plantlets, and low or no expression in other tissues. The expression patterns of *StCOL2* and *StCOL9* were similar, with the highest expression levels in flowers, followed by the shoots and leaves of plantlets. The expression patterns of *StCOL3* and *StCOL4* were similar, with the highest expression levels in the shoots and leaves of plantlets, and low or no expression in the other tissues. The expression of *StCOL5* was higher than other tissues in the leaves of plantlets. *StCOL7* was expressed in all tissues and highly expressed in flowers, the leaves of plantlets and the roots. *StCOL8* was highly expressed only in the leaves of plantlets, but not expressed in the other tissues. *StCOL12* was highly expressed in roots. The expression patterns of *StCOL6*, *StCOL10* and *StCOL14* were similar, with the highest expression in flowers and low or almost no expression in the other tissues. In conclusion, most *StCOL* genes were highly expressed in the leaves of plantlets, while *StCOL8* was specifically expressed only in the leaves of plantlets. The analysis of the 2000-bp sequence upstream of the start codon of each *StCOL* family member revealed several regulatory elements related to the induction of phytohormones, such as gibberellin, abscisic acid, auxin and methyl jasmonate. In addition, the upstream regions of some genes also contained regulatory elements and transcription factor binding sites related to light, drought and low temperature (Appendix A).

## 4. Discussion

The *CONSTANS*-like gene family is involved in regulating the tuber formation and development of potato [7,9]. The genome-wide identification and functional analysis of the *COL* gene family will help to determine the regulatory role of *COL* in potato tuber formation and development and the regulatory mechanism of its response to the photoperiod.

### 4.1. Phylogenetic and Gene Structure Analysis

Researchers have identified the *COL* gene families of many plants, such as *Arabidopsis* [61], *Petunia* [12], *Fragaria ananassa* [62], *Capsicum* [63] and *Cannabis* [64]. However, the systematic identification of *COL* genes throughout the whole genome of potato has not been reported in detail. In our study, we identified 14 potato *COL* gene family members at the genome-wide level (Table 1). Blast-2.13.0 + analysis showed that the similarity between the *StCOL1* sequence and *StCO* sequence reported by González-Schain et al. was 98.148% [8]. We concluded that *StCOL1* might be the *StCO* gene. *StCOL* genes are distributed on eight chromosomes (Appendix A), with differences in the number of *COL* family members in different plants: 17 in *Arabidopsis* [61], 33 in *Petunia* [12], 13 in *Cannabis* [64], 10 in *F. ananassa* [62] and 10 in *Capsicum* [63]. The number of gene family members has shrunk or expanded in different species, which might be caused by an adaptation to the different environmental conditions during the long-term evolution of a plant species.

*COL* family members have three conserved domains: two B-box domains and a CCT domain [13]. The evolution rate of plant regulatory genes is fast, and the evolution rate of different domains of *COL* family proteins is substantially different [18]. Although COL proteins in both subgroups I and III contain B-box domains, these domains are substantially different. The B-box domains of groups I and III are absent in some dicotyledons [65], but this deletion was not observed in potato. Potato group I contains five *StCOL* genes, group II contains three *StCOL* genes and group III contains seven *StCOL* genes. Subgroup III contains the largest number of genes. Subgroup I and III proteins had motif 1, while subgroup II proteins had not; subgroup I and III proteins contained Bbox_SF1 and Bbox_SF2, subgroup II proteins contained Bbox_SF1 (Figure 2). The *StCOL* gene structure of groups I and III is relatively conserved, while the *StCOL* gene structure of group II is considerably differentiated, indicating that the expansion of the *COL* gene family has mainly occurred in subgroup II.

A gene structure analysis of potato *COL* gene family members revealed that the exon–intron length and structure within the same subgroup are similar, and the COL proteins in the same subgroup have highly similar motif structures, indicating that the exon length and number and the position of the introns of *COL* genes located in the same subgroup have been relatively conserved during the long-term evolution of potato. This suggests that *COL* family genes may have the same fixed splicing pattern [66]. In group III, the *StCOL7*, *StCOL8*, *StCOL10* and *StCOL14* genes were different in structure, but the amino acid sequences encoded were similar. Such a variation may result from exon duplication, variation and recombination during the genome evolution, or from intron insertion and loss [62].

Our phylogenetic analysis showed that the COL proteins of potato and tomato are closely related and share high sequence similarity, which was similar to the research results of Huang et al. in pepper [63]. A collinearity analysis of *S. tuberosum* and *Arabidopsis*, *S. lycopersicum* and *N. tabacum* genomes showed that *StCOL* has the closest relationship with *S. lycopersicum* and *N. tabacum* orthologs. This result is consistent with the taxonomic relationships of potato, tomato and tobacco within the *Solanaceae*.

### 4.2. Functional Conservation and Differentiation of COL Gene Family Members in S. tuberosum

*COL* genes affect the regulation of growth and development of different tissues and organs of plants, and show obvious tissue expression specificity [11,67,68,69]. The tissue expression data showed that potato *COL* family genes are differentially expressed in different tissues (leaves, stems, tubers, roots, flowers, shoots, leaves of plantlets, stems of plantlets, microtubers). Most *StCOL* genes are highly expressed in the leaves of plantlets (Figure 5), which is consistent with findings in other species [38,39,70]. Since leaves are light-sensing organs in plants, this result suggests that *StCOL* may be involved in photoperiodic sensitivity. The photoperiod regulation of tuber formation and plant flowering have common factors [3]; leaves perceive both photoperiod signals and the duration of sunlight to induce flowering and tuber formation. *StCOL1*, *StCOL2*, *StCOL3*, *StCOL4*, *StCOL5*, *StCOL7*, *StCOL8*, *StCOL11* and *StCOL13* are highly expressed in the leaves of plantlets. After potato leaves perceive light signals, these genes may be highly expressed in leaves to regulate tuber formation and development.

*StCOL7* and *StCOL12* are highly expressed in roots, and *StCOL7* is also highly expressed in other tissues, suggesting that *StCOL7* may play a role in many other aspects of potato development. By contrast, the expression of different *StCOL* genes can be detected in the same tissue, suggesting their functional redundancy. The *AtCOL12* gene affects the regulation of flowering time and morphogenesis in *Arabidopsis* [71], and *StCOL6*, *StCOL10* and *StCOL14* were highly expressed in flowers in our study. These genes may therefore be involved in the regulation of flowering in the potato. Expression of *NnCOL5* is positively correlated with the expansion of *Nelumbo nucifera*. Heterologous *NnCOL5* expression in the potato increases tuber mass and starch content without changing the number of tubers under SD conditions [70]. In our study, *StCOL7* was highly expressed in tubers and microtubers (Figure 5), indicating that it may be involved in the regulation of tuber expansion. Studies have found that *StCOL1* does not directly regulate the tuber signaling of *FT* homolog *StSP6A*, but activates the FT-like *StSP5G* homolog, which inhibits tuber formation. The silencing of *StCOL1* or *StSP5G* leads to the activation of *StSP6A* in leaves, and negative regulation of tuber formation by StCOL1 depends on the *StSP5G* gene [9]. An additional level of control influenced by the expression of *StCOL1*, *StSP5G* and *StSP6A* determines whether the plant generates tubers [41,42]. Suppression of *SlCOL1* expression in tomato leading to the promotion of fruit development was found [72]. In this study, *StCOL13* was not expressed in tubers, but expressed at low levels in microtubers, and showed high expression levels in leaves. This may be because high expression levels of *StCOL13* in leaves promote the expression of the *StSP5G* gene or inhibit the expression of *StSP6A*, which then regulates the expression of *StCOL13* in tubers, thereby inhibiting the formation and development of tubers.

The analysis of the promoter regions of *StCOL* genes showed that these promoter regions contain not only light-responsive elements, but also a variety of regulatory phytohormone induction elements and stress-responsive elements (Appendix A). The expression of *StCOL* genes may be related to light, phytohormones and resistance to abiotic stress. *COL* homologs, such as *AtCOL7*, have dual roles, promoting branching under high R:FR conditions but enhancing the shade avoidance syndrome under low R:FR conditions [68]. The expression of *MaCOL1* in peel changes slightly, while the accumulation of *MaCOL1* transcripts in the pulp obviously increases during natural or ethylene-induced fruit ripening in banana [67]. The transcript levels of other ABA biosynthesis and stress-related genes show an enhanced induction in *AtCOL4*-overexpressing and WT plants, but not in the *atcol4* mutant in the presence of ABA or salt stress. Thus, *AtCOL4* is involved in ABA and salt-stress responses through the ABA-dependent signaling pathway [73]. The COL protein OsCOL9 interacts with Receptor for Activated C-Kinase 1 (OsRACK1) and enhances rice blast resistance through the salicylic acid and ethylene signaling pathways [74]. The *OsGhd2* gene controlling the grain number, plant height and heading date plays an important role in accelerating the drought-induced senescence of leaves in rice [75]. *BnCOL2* regulates plant tolerance to drought stress by regulating the expression of ABA-responsive genes and drought stress-related genes in *Arabidopsis* [11]. The CO-High Expression of Osmotically Responsive Gene 1 (CO-HOS1) module in *Arabidopsis* is crucial for fine tuning the flowering response to the photoperiod in response to short-term temperature fluctuations [76]. All of these reports indicate that *COL* genes regulate growth and development in plants in response to light, phytohormones, stress and other signals, thus helping plants to better adapt to their environments. However, the possible important functions of these genes in regulating flowering and tuber formation and development in potato need to be further studied.

## 5. Conclusions

We identified 14 *CO*-like genes from the potato genome, which belong to three subgroups and are unevenly distributed on eight chromosomes. The COL proteins of potato and tomato are closely related in the evolutionary tree and show a high sequence similarity. There are 17 orthologous gene pairs between *S*. *tuberosum* and *S. lycopersicum*. The evolution rate of *COL* homologs is controlled by purification selection in *Arabidopsis*, *S*. *tuberosum* and *S. lycopersicum*. Tissue-specific expression characteristics suggest a functional differentiation of *StCOL* genes during evolution. The promoter regions of *StCOL* genes contain many regulatory elements for phytohormone, light and stress signals. Although the specific functions and regulatory mechanisms of *COL* genes in potato were not thoroughly explored in this study, our finding provides a reference for further exploring the functions of *COL* gene family members in regulating tuber formation and development of potato.

## Figures and Tables

**Figure 1 genes-14-01174-f001:**
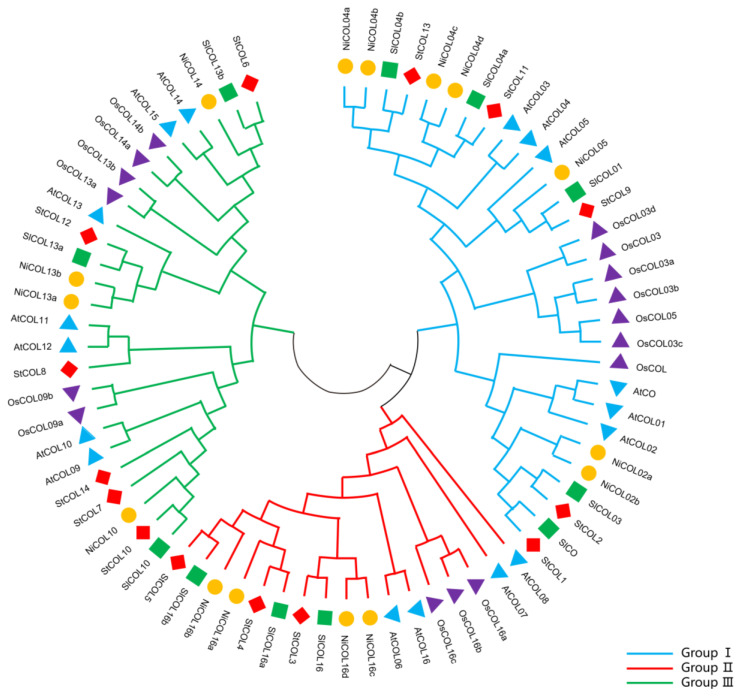
A neighbor-joining phylogenetic tree of COL family members from *S*. *tuberosum* (St), *S*. *lycopersicum* (Sl), *N*. *tabacum* (Ni), *O. sativa* (Os) and *A. thaliana* (At). The number on the branches represents the reliability percent of the bootstrap values based on 1000 replications.

**Figure 2 genes-14-01174-f002:**
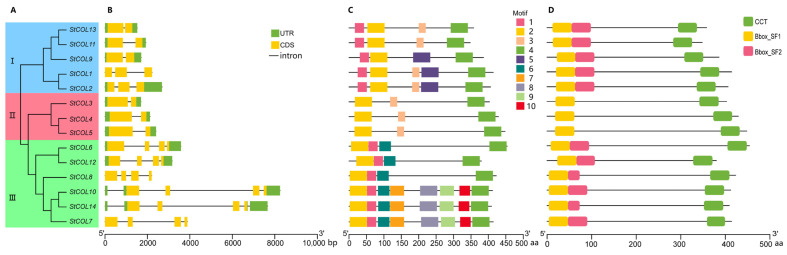
Motif distributions and exon-intron structures of COL family members in *S. tuberosum*. (**A**) *COL* phylogenetic tree of *S. tuberosum*; (**B**) exon–intron distribution of *COL* genes; (**C**,**D**) distribution characteristics of the conserved motifs of COL proteins; UTR, untranslated region; CDS, coding sequence.

**Figure 3 genes-14-01174-f003:**
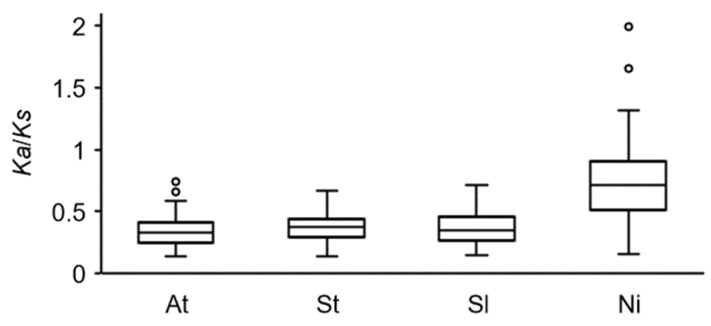
Statistics of the selection pressures on collinear *COL* genes within species. At, *A. thaliana;* St, *S. tuberosum*; Sl, *S. lycopersicum*; Ni, *N. tabacum*.

**Figure 4 genes-14-01174-f004:**
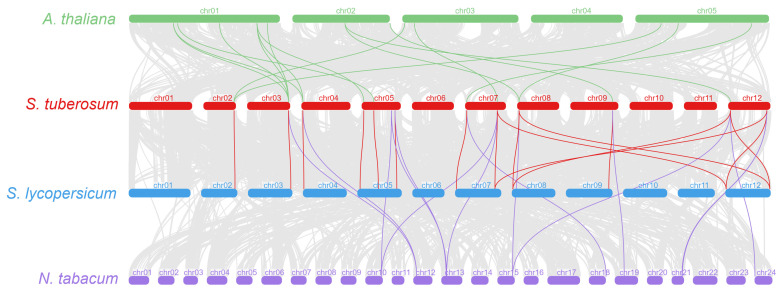
Syntenic relationships of *COL* genes in *A. thaliana*, *S*. *tuberosum*, *S*. *lycopersicum* and *N*. *tabacum*. Gray lines in the background indicate collinear blocks between *S. tuberosum* and other genomes, while red, green and purple lines highlight syntenic *COL* gene pairs. The chromosome number is indicated at the top of every chromosome.

**Figure 5 genes-14-01174-f005:**
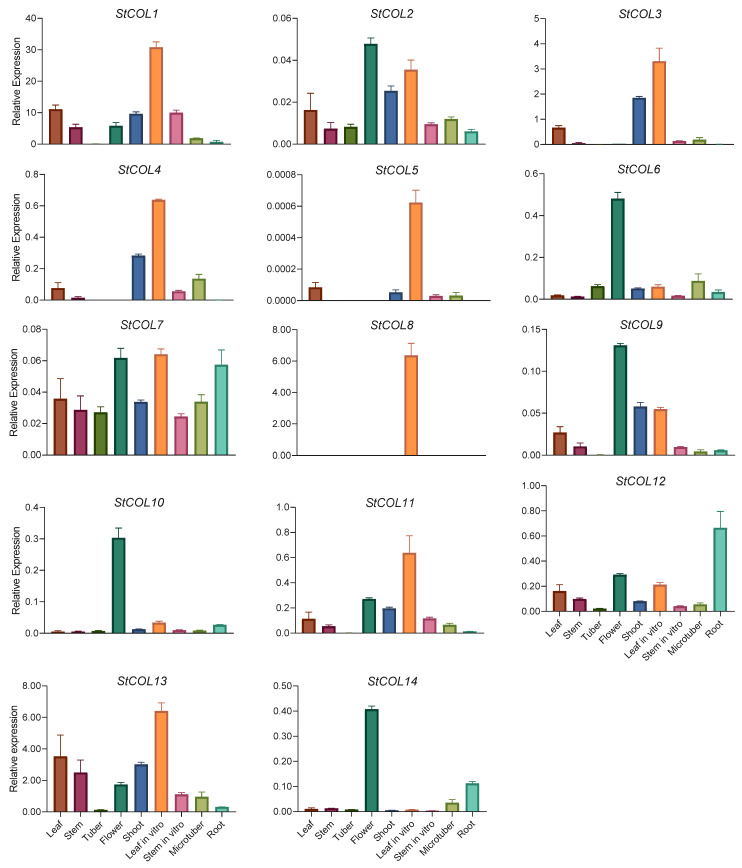
Expression patterns of the 14 *StCOL* genes in nine different tissues determined using qPCR. The data points represent mean ± SD.

**Table 1 genes-14-01174-t001:** Profiles of *COL* gene family members identified in *S. tuberosum*.

Gene	Gene ID	Amino Acid Number/aa	Molecular Weight/Da	Theoretical Isoelectric Point	Subcellular Localization
*StCOL1*	Soltu.DM.02G030260.1	413	45,873.9	5.82	Nucleus
*StCOL2*	Soltu.DM.02G030280.1	405	44,940.9	5.57	Nucleus
*StCOL3*	Soltu.DM.03G034030.1	401	45,829.1	5.61	Nucleus
*StCOL4*	Soltu.DM.04G002000.1	428	48,809.4	5.44	Nucleus
*StCOL5*	Soltu.DM.05G003170.1	447	51,218.4	5.62	Nucleus
*StCOL6*	Soltu.DM.05G012520.1	453	49,919.0	6.09	Nucleus
*StCOL7*	Soltu.DM.05G017200.1	413	45,047.1	5.61	Nucleus
*StCOL8*	Soltu.DM.05G018860.1	420	46,241.8	5.20	Nucleus
*StCOL9*	Soltu.DM.07G001900.1	385	42,403.6	6.05	Nucleus
*StCOL10*	Soltu.DM.07G014960.1	411	45,673.9	5.41	Nucleus
*StCOL11*	Soltu.DM.08G002010.1	347	38,651.0	5.30	Nucleus
*StCOL12*	Soltu.DM.09G022940.1	379	43,059.9	5.78	Nucleus
*StCOL13*	Soltu.DM.12G003910.1	357	39,093.3	5.30	Nucleus
*StCOL14*	Soltu.DM.12G023800.1	411	45,673.9	5.41	Nucleus

## Data Availability

The genome sequence, proteins and corresponding coding sequences of *S. tuberosum* were downloaded from the Phytozome v13 website (https://phytozome.jgi.doe.gov/pz/portal.html, accessed on 10 November 2022). COL family protein sequences of *A. thaliana*, *S. lycopersicum*, *N. tabacum* and *O. sativa* were download from NCBI (https://www.ncbi.nlm.nih.gov/, accessed on 10 November 2022).

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
