# Peer review of "Genome-Wide Identification, Characterization and Expression Profiling of the CONSTANS-like Genes in Potato (Solanum tuberosum L.)"

_genes, 2023, doi:10.3390/genes14061174_

Round 1

Reviewer 1 Report

This paper presents an overview of the CONSTANS-LIKE genes in potato. It would be of moderate interest to those interested in tuberization in potatoes. Key findings include a blast search to identify candidate COL genes, phylogenetic grouping, characterization of the putative domain structure of the proteins, gene expression measured by qRT-PCR, and a motif search of the proximal promoter regions. The authors have made some effort to hypothesize about the functions of specific alleles based on these results, however, because of a serious flaw in the methodology, some of these conclusions may be invalid. The critical error is that the primers listed for each StCOL gene model amplify a different StCOL gene indicating that the gene models were mixed up somewhere along the way. See specific comments below. Extensive edits would needed to correct this error since the main findings of the paper depend on these erroneous data.

Specific comments by line.

15: See the comment regarding Table 1. There are only 14 genes.

93: Be more specific about what these breakthroughs are.

101: bioinformatics methods is too vague.

116: Inconsistent font.

124: Additional details should be provided regarding the age, maturity, stage, etc. of each of the tissues, particularly the size and maturity of the tubers.

Table 1: StCOL14 and StCOL15 are different transcript isoforms of the same gene and likely do not represent truly independent proteins. The authors should identify which transcript is representative and confine their analysis to that isoform.

Figure 3: The label for this graph is not in English.

Figure 4. This figure would be more visually appealing if the bars were solid rather than having a white ring around the edge.

Figure 5 and Supplemental Table 1: As stated in the opening paragraph, every primer listed in Supplemental Table 1 blasts to the wrong gene. The primers listed for StCOL2 and StCOL15 both blast to StCOL4. The similarity between the graphs for StCOL2 and StCOL15 reflect this. Conversely the graphs for StCOL14 and StCOL15 differ significantly despite those merely being alternative transcript isoforms because those primers map to StCOL10 and StCOL4, respectively. The primers listed for StCOL7 map to both StCOL1 and StCOL2 equally well. Alternative primers should be chosen to distinguish StCOL1 and StCOL2. The qRT-PCR amplicons should be sequenced to ensure the correct gene is the only gene being amplified. If the lab has the expertise, uniquely mapping RNA-seq reads should be compared for each StCOL gene between published RNA-Seq datasets from various tissues.

Reviewer 2 Report

The article "Genome-Wide Identification, Characterization, and Expression 2 Profiling of the CONSTANS-like Genes in potato (Solanum tu-3 berosum L.)" by Li and co-workers.

Add a few latest references in the Introduction section of the manuscript.

Line no 145 write: Phylogenetic tree reconstruction, gene structure and protein motifs

Use more bioinformatics software to cross-validate

 Figure 3: correctly mentioned the Axis value in English

Figure/Fig. Correct the whole Ms

Kindly check the grammatical mistake throughout the Ms.

1.   What is the main question addressed by the research?

Response

Expression specificity of potato COL genes in different tissues development stages.

2.   Do you consider the topic original or relevant in the field? Does it address a specific gap in the field?

Response

Yes

3. What does it add to the subject area compared with other published material?

Response

Cite some few research paper i.e., doi.org/10.3390/genes14020468, DOI: 10.3390/ijms222413535, DOI: 10.3390/genes14020468, doi.org/10.1186/s12864-019-5698-x, doi.org/10.3389/fpls.2021.804600, doi.org/10.1186/s12864-022-09037-y

4. What specific improvements should the authors consider regarding the methodology? What further controls should be considered?

Response

In methodology part, the authors add few more bioinformatics software to validate.

5. Are the conclusions consistent with the evidence and arguments presented and do they address the main question posed?

Response

Conclusion part is satisfactory.

6. Are the references appropriate?

Response

Yes

7. Please include any additional comments on the tables and figures.

Response

Tables and figures are perfect in the present form.

Round 2

Reviewer 1 Report

While the largest error in the qRT-PCR experiment has been corrected, little improvement was attempted from the previous version, and many of my concerns were not addressed.

StCOL1 primers still equally amplify StCOL1 and StCOL2 equally well. PCR amplicons were not sequenced to verify amplification of the correct gene. No RNA-seq data is included or referenced. Throughout the methods section, experiments and bioinformatic analyses are inadequately described.

97: It is not adequate to say “a series of breakthroughs” and then only list one.

271: The tissues are still not adequately described

346: More details are necessary here.

Figure S4: The authors completely changed this figure from the previous version enlarging all the motifs and changing the colors. The distribution is also completely different. Given the lack of details regarding this analysis, it is impossible to tell whether these representations are accurate. It seems unlikely that conserved motifs would be as large as those depicted here. Is the scale correct? Perhaps this depicts a region far shorted than 2kb.
